# Layer 6 corticocortical neurons are a major route for intra- and interhemispheric feedback

**Simon Weiler[1], Manuel Teichert[2†], Troy W Margrie[1]\*†**

[1]Sainsbury Wellcome Centre for Neuronal Circuits and Behavior, University College London, London, United Kingdom; [2]Jena University Hospital, Department of Neurology, Jena, Germany

## eLife Assessment

This **important** study compares the cortical projections to primary motor and sensory areas originating from the ipsilateral and contralateral hemispheres. Results show that, while there is substantial symmetry between the two hemispheres regarding the areas sending projections to these primary cortical areas, contra-hemispheric projections had more inputs from layer 6 neurons than ipsi-projecting ones. The evidence is **compelling** and the conclusions are supported by rigorous analyses.

**\*For correspondence:**
t.margrie@ucl.ac.uk

†These authors contributed equally to this work

**Competing interest:** The authors declare that no competing interests exist.

**Abstract** The neocortex comprises anatomically discrete yet interconnected areas that are symmetrically located across the two hemispheres. Determining the logic of these macrocircuits is necessary for understanding high level brain function. Here in mice, we have mapped the areal and laminar organization of the ipsi- and contralateral cortical projection onto the primary visual, somatosensory, and motor cortices. We find that although the ipsilateral hemisphere is the primary source of cortical input, there is substantial contralateral symmetry regarding the relative contribution and areal identity of input. Laminar analysis of these input areas show that excitatory Layer 6 corticocortical cells (L6 CCs) are a major projection pathway within and between the two hemispheres. Analysis of the relative contribution of inputs from supra- (feedforward) and infragranular (feedback) layers reveals that contra-hemispheric projections reflect a dominant feedback organization compared to their ipsi-cortical counterpart. The magnitude of the interhemispheric difference in hierarchy was largest for sensory and motor projection areas compared to frontal, medial, or lateral brain areas due to a proportional increase in input from L6 neurons. L6 CCs therefore not only mediate long-range cortical communication but also reflect its inherent feedback organization.

## Introduction

The cortex is divided into two hemispheres that do not function independently. Rather, they are connected by commissural fibers, most notably the corpus callosum which is thought to coordinate function across the brain (*Gazzaniga, 2000*; *Innocenti et al., 2022*; *Ocklenburg and Guo, 2024*; *Zhou et al., 2013*). Within these circuits feedforward and feedback pathways represent two fundamental types of information flow (*Berezovskii et al., 2011*; *Markov and Kennedy, 2013b*). In the case of sensory processing, feedforward connections typically carry sensory input toward areas higher in cortical hierarchy where more complex processing occurs (*Felleman and Van Essen, 1991*; *Mumford, 1992*; *Rockland and Pandya, 1979*; *Siegle et al., 2021*). Conversely, cortical feedback typically originates in higher-order areas and projects back to brain areas of lower hierarchical rank (*Felleman and*

*Van Essen, 1991*; *Markov et al., 2014*; *Rockland and Pandya, 1979*). These pathways are believed to be crucial for modulating sensory processing (*Weiler et al., 2024*), enhancing signal fidelity (*Xu et al., 2012*), integrating contextual information (*Zhang et al., 2014*), and forming predictions (*Keller and Mrsic-Flogel, 2018*; *Leinweber et al., 2017*; *Rao and Ballard, 1999*).

Based on its cytoarchitecture the mammalian neocortex contains up to six layers that are believed to play distinct roles in feedforward and feedback processing (*Harris et al., 2019*; *Markov et al., 2013a*; *Markov et al., 2014*). For instance, retrograde tracing (*Markov et al., 2014*) and physiological analysis (*Bastos et al., 2015*; *Buffalo et al., 2011*) in the visual cortical system of primates indicate that supragranular layers are the main source of feedforward projections while infragranular layers are the main source of feedback input. Moreover, the ratio of supra- to infragranular layer projections has been used to reveal the anatomical basis of visual cortical hierarchy in primates and mice (*Barone et al., 2000*; *D'Souza et al., 2022*; *Markov et al., 2014*; *Yao et al., 2023*). While it is established that functional cortical hierarchy (at least in the visual system) reflects the laminar organization of cortical input within the ipsilateral hemisphere, it remains unknown how this is combined or even reflects the organization of input from the contralateral hemisphere. More specifically, to date there remains no cortex-wide anatomical analysis that investigates the nature of circuit hierarchy onto a given area that arises from both hemispheres. Also, which cortical layers are responsible for the establishment of the intra- versus the interhemispheric cortical hierarchy is not understood.

In this study, we utilized state of the art retro-AAV-based neuronal tracing in adult mice to determine the anatomical location of input neurons and map the cortex-wide projectome onto the primary visual (VISp), the primary somatosensory barrel field (SSp-bfd), and the primary motor (MOp) cortices. We employ an anatomically based hierarchy metric that not only enables us to define the contribution of Layer 2/3 (L2/3) and the infragranular layers but also enables us to disassociate the roles of Layer 5 (L5) and L6 in the organization of cortical feedback circuits. Our results indicate extensive and symmetric projection patterns across the two hemispheres and reveal a key role for L6 in routing cortical feedback information.

## Results

To systematically map the cortical projections onto primary sensory and motor cortical areas in adult mice, we utilized retrograde tracing with a recombinant AAV-variant, AAV-EF1a-H2B-EGFP (nuclear retro-AAV), which is taken up by axon terminals (*Tervo et al., 2016*) and results in EGFP expression in the nuclei of projection neurons (*Figure 1A*, *Figure 1—figure supplement 1A*). This tracer was injected in either VISp (*n* = 6), SSp-bfd (*n* = 6), or MOp (*n* = 6), spanning across all cortical layers of each target area (*Figure 1A*, *Figure 1—figure supplement 2*, *Figure 1—figure supplement 3*). Importantly, only brains that had viral transduction levels of the white matter below 0.1% of the total bolus volume were included (*Figure 1—figure supplement 2B*). Following ex vivo two-photon tomography (*Ragan et al., 2012*) and 3D brain registration (*Niedworok et al., 2016*) detected cell nuclei (*Tyson et al., 2021*) were assigned according to the cortical areas of the Allen Mouse Brain Common Coordinate Framework (*Claudi et al., 2020*; *Wang et al., 2020*, CCFv3, *Figure 1B*, excluding the targeted injection area). We found that the vast majority of neurons (>99%) projecting to all target areas were non-overlapping with GAD-expressing cells (*Figure 1—figure supplement 4A*). Regarding L6 projection neurons, they were also found to be non-overlapping with NTSR1-expressing L6 corticothalamic cells and therefore indicative of Layer 6 corticocortical cells (L6 CCs) (*Figure 1—figure supplement 4B, C*).

### The areal and modular organization of cortical input onto VISp, SSp-bfd, and MOp

The resultant projection maps revealed labeling of several hundred thousand neurons located throughout both cortical hemispheres (*Figure 1—figure supplement 1B*), with approximately 80% located ipsilaterally and 20% contralaterally (*Figure 1B, C*, p < 0.05, two-sided Wilcoxon signed-rank test), irrespective of identity of the target injection site. While all three target areas received ipsilateral input from almost all cortical areas (VISp, range: 41–44, *n* = 6; SSp-bfd, range: 43–44, *n* = 6, MOp, range: 33–39, *n* = 6, *Figure 1D*) the two sensory cortical targets received input on average from more areas than the primary motor cortex (*Figure 1D*, sensory = 43.58 ± 0.26, *n* = 12 vs. motor 36.67 ± 0.99, *n* = 6; p < 0.001, two-sided Wilcoxon rank-sum test). Comparing the inputs from both

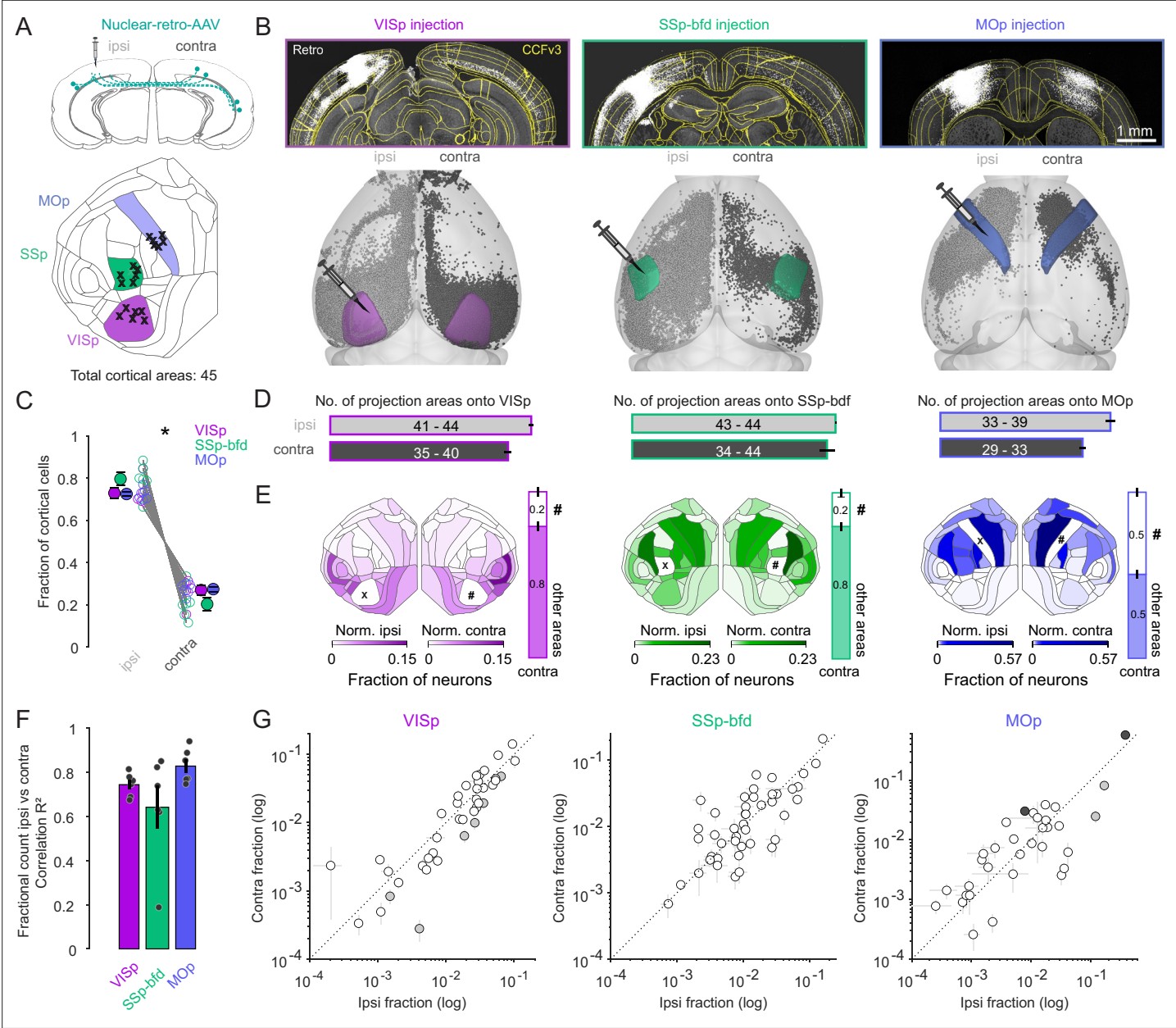

**Figure 1.** Widespread and bilaterally symmetrical cortical projections onto VISp, SSp-bfd, and MOp. (**A**) Retrograde nuclear retro-AAV was injected across all layers of the primary visual cortex (VISp), somatosensory barrel field (SSp-bfd), and the motor cortex (MOp) of mice. The locations of the centroids of the injection bolus is shown on the CCFv3 cortical flat map. Injections performed in the right hemisphere (*n* = 11) are flipped to their corresponding location in the left hemisphere. (**B**) Two photon images showing examples of injection sites and labeled nuclei of retro-AAV labeled cells. Visualization of detected projection neurons for three individual mice warped into the 3D rendered space of the two hemispheres. (**C**) Plot showing the fraction of cortical cells per hemisphere detected for each mouse and pooled according to each injection site (mean ± SEM). Asterisks indicate statistical significance (p < 0.05). (**D**) Bar graphs indicating the mean number (± SEM) of cortical areas in which labeled neurons were observed. Text indicates the range of areas. (**E**) Flat maps showing the relative cell counts normalized for each hemisphere excluding both the injection target (X) area its homotopic counterpart (#). Right, bar graphs indicating the fraction of cells in the contralateral hemisphere located in the homotopic versus all other cortical areas (mean ± SEM). Note that areas ILA (5), ENTl (44), and ENTm (45) are not represented in the cortical flat map on the left (see also Figure S1C). (**F**) Bar graphs showing the average correlation ($R^2$ ± SEM) of the ipsi- versus contralateral fractional count superimposed on the individual $R^2$ values for each experiment. (**G**) Plot of the ipsi- versus contralateral fractional count for each of the three target areas. Each circle represents the average of six fractional count values (± SEM) for a given area. Open circles indicate instances where the difference of the ipsilateral values are not significant from the contralateral values. Black circles indicate a significant contralateral bias while gray circles indicate an ipsilateral bias. Dashed line indicates the unity line.

The online version of this article includes the following source data and figure supplement(s) for figure 1:

*Figure 1 continued on next page*

*Figure 1 continued*

**Source data 1.** Data corresponding to panels A, B, C, D, E, F, and G.

**Figure supplement 1.** Experimental workflow and cell detection.

**Figure supplement 1—source data 1.** Data corresponding to panel B.

**Figure supplement 2.** Target area viral expression quantification.

**Figure supplement 2—source data 1.** Data corresponding to panels B, C and D.

**Figure supplement 3.** Latero-medial view of target area viral expression.

**Figure supplement 3—source data 1.** Data corresponding to panel B.

**Figure supplement 4.** Molecular identity of long-range projection neurons.

**Figure supplement 4—source data 1.** Data corresponding to panel B.

hemispheres showed that VISp and MOp injections revealed partial asymmetry in the number of areas labeled per hemisphere (*Figure 1D*, VISp ipsilateral = 43.33 ± 0.49 vs. VISp contralateral 38.33 ± 0.72, MOp ipsilateral = 36.67 ± 0.99 vs. MOp contralateral 30.67 ± 0.67, p < 0.05, signed-rank test), whereas the source of input to SSp-bfd appeared to be mirrored across the two hemispheres (*Figure 1D*, SSp-bfd ipsilateral = 43.83 ± 0.17 vs. SSp-bfd contralateral 42 ± 1.61, p = 0.5, two-sided Wilcoxon signed-rank test).

Importantly, we found there to be no instance where a projection area in the contralateral hemisphere did not have an ipsilateral counterpart. These data show that both primary sensory and motor cortices receive an abundance of functionally diverse input from cortical areas outside their primary contralateral (homotopic) area indicating significant cross-modal integration within and between the two hemispheres (*Figure 1E*).

To understand the relative contribution of a given area to the projectome for each hemisphere we performed a correlation analysis on the relative fraction of labeled cells per area normalized to the total hemispheric count. Excluding the homotopic target area, that always contained the highest number of cells per area, but including all the cases where the number of cells in an area exceed ten in at least three mice (see Methods), we found a strong correlation between the fractional count of neurons in a given ipsilateral brain area with its contralateral counterpart (VISp $R^2$ = 0.74 ± 0.02, p < 0.05; SSp-bfd $R^2$ = 0.64 ± 0.1, p < 0.05; MOp $R^2$ = 0.83 ± 0.03, p < 0.05, linear regression fit, *Figure 1E, F*). This relationship was observed independent of the target identity (*Figure 1F, G*). When directly comparing the within hemisphere relative projection weights of the individual cortical areas across the two hemispheres, we found that 33/40 (VISp), 44/44 (SSp-bfd), and 29/33 (MOp) were not significantly different from one another which indicates significant bilateral symmetry in the relative contribution of a given area to its hemispheric projectome (*Figure 1G*, individual one-sample *t*-tests on contra–ipsi difference, Bonferroni multiple comparison correction).

Based on their degree of inter-areal cortical connectivity it has been suggested that the mouse neocortex consists of six anatomical modules referred to as prefrontal, lateral, somatomotor, visual, medial, and auditory (*Figure 2A*, *Harris et al., 2019*). To determine whether the areal projections reflect any underlying modular organization we next analyzed the distribution of projection neurons according to their respective module. First, while the VISp projection was heavily dominated by input from the visual module (*Figure 2B*, ipsilateral visual module fractional count 0.42 ± 0.02 vs. lateral module 0.19 ± 0.02; contralateral visual module 0.42 ± 0.03 vs. lateral module 0.22 ± 0.02, p < 0.001, two-sided unpaired *t*-test), both the SSp-bfd and MOp targets received input predominantly from the somatomotor module (*Figure 2B*, SSp-bfd: ipsilateral somatomotor module fractional count 0.62 ± 0.05 vs. visual module 0.16 ± 0.03, contralateral somatomotor module 0.53 ± 0.02 vs. lateral module 0.24 ± 0.02, p < 0.001; MOp: ipsilateral somatomotor module 0.83 ± 0.02 vs. lateral module 0.08 ± 0.01, contralateral somatomotor module 0.71 ± 0.03 vs. lateral module 0.14 ± 0.01, p < 0.001, two-sided unpaired *t*-test). For both the visual module projection onto VISp and the somatomotor module projections to SSp-bfd and MOp there were similar contributions from the majority of the underlying areas (*Figure 2C*). Second, although both VISp and SSp-bfd received significant input from all modules in both hemispheres, we find there to be almost no input onto MOp from the visual, auditory, and medial modules in either hemisphere (*Figure 2B*, less than 0.01 fractional count for each of these three modules). Finally, there also appears to be biases in the relative density of

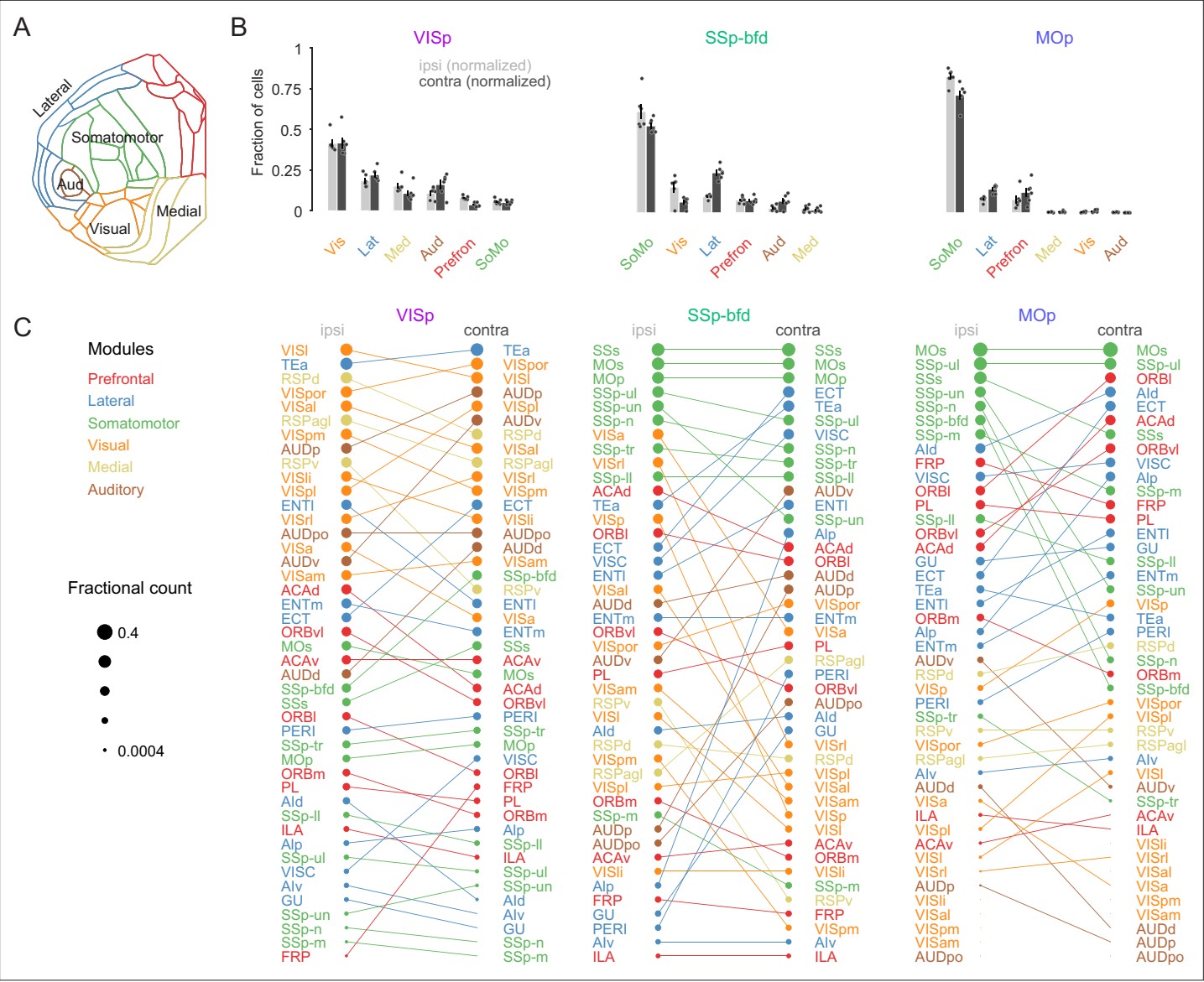

**Figure 2.** Modular organization of the intra- and interhemispheric cortical projection onto VISp, SSp-bfd, and MOp. (**A**) Schematic of the cortical flat map highlighting the six cortical modules and their corresponding areas. (**B**) Bar graphs of the normalized fractional cell counts for each of the six modules for ipsi- and contralateral hemispheres ranked by decreasing ipsilateral weights. For the contralateral hemisphere the homotopic target region is not included. (**C**) Rank plot based on the relative weights of connectivity onto the target areas VISp, SSp-bfd, and MOp for all cortical brain areas. The size of the circles indicates relative fractional count on a logarithmic scale. Corresponding areas on the ipsi- and contralateral hemisphere are connected with a line.

The online version of this article includes the following source data for figure 2:

**Source data 1.** Data corresponding to panels B and C.

projections. For example, we find that the lateral module in the contralateral hemisphere projecting to VISp is more strongly weighted than its ipsilateral counterpart and that this is driven primarily by the connection from the temporal association area (TEa) and ectorhinal cortex (ECT) (***Figure 2B, C***, lateral module ipsilateral 0.19 ± 0.02 vs. contralateral 0.22 ± 0.02, p = 0.05, one-tailed paired *t*-test). While this contralateral dominance of the lateral module also emerges from the projection profiles to SSp-bfd and MOp it is rather mediated by many, albeit less dominant, lateral module areas (***Figure 2C***, SSp-bfd: lateral module fractional count ipsilateral 0.1 ± 0.01 vs. contralateral 0.24 ± 0.02, p < 0.001, one-tailed paired *t*-test, MOp: lateral module ipsilateral 0.08 ± 0.01 vs. contralateral 0.14 ± 0.01, p < 0.01, one-tailed paired *t*-test).

## The laminar organization of cortical input to VISp, SSp-bfd, and MOp

Next, we asked whether there were specific subsets of projection neurons that might differentially contribute to the local and global projection by quantifying the distribution of neurons across all cortical layers in both hemispheres (*Figure 3A1–A4*). First, for each injection target we find that the majority of detected neurons were located in L2/3, L5, and L6a (*Figure 3B*) with a very low fraction of labeled neurons in L4 and L6b. To begin to explore the laminar structure in the input to the three target areas we next compared the relative dominance of these three main layers. Regardless of hemisphere, L2/3 was found not to be the dominant layer of inter-areal input (*Figure 3C1, C2, D1, D2*). For VISp and MOp L6 dominated both the intra- and interhemispheric projection (*Figure 3C1, C2, D1, D2*, p < 0.01, one-way ANOVA, Tukey–Kramer multiple comparison correction). In the case of SSp-bfd, L5 and L6 dominated the contralateral input (*Figure 3C2, D2, L5* vs. L2/3 and L6 vs. L2/3, p < 0.01, one-way ANOVA, Tukey–Kramer multiple comparison correction) and shared dominance with L2/3 only for the ipsilateral projection (*Figure 3C1, D1, L5* vs. L2/3, p = 0.06; L6 vs. L2/3, p = 0.64, one-way ANOVA, Tukey–Kramer multiple comparison correction). When pooled across target areas, we find that the majority of projection source areas display L6 dominance, followed by L5 with only a small fraction of areas exhibiting L2/3 dominance (*Figure 3E*, p < 0.05, one-way ANOVA, Tukey–Kramer multiple comparison correction). These data show that L6 CCs are a key player mediating intra- and interhemispheric connectivity onto primary sensory and motor areas.

## Cortical input projections onto VISp, SSp-bfd, and MOp reflect a hierarchical organization

In primates, the laminar distribution of projection neurons has been used to interpret the degree to which corticocortical networks might be feedforward or feedback (*Barone et al., 2000*; *Markov et al., 2014*; *Vezoli et al., 2021*). A predominance of incoming projection neurons from supragranular layers (L2/3) signifies a feedforward connection, whereas a higher proportion in infragranular layers (Layers 5 and 6) is considered to reflect a feedback projection (*Figure 4A*). Additionally, cortical areas with a higher proportion of projection neurons in infragranular, compared to supragranular layers, are suggested to have a higher hierarchical rank (*Figure 4A*). Thus, the hierarchical rank of a specific projection area can be estimated by calculating the fraction of infragranular labeled neurons (fILN) (*Figure 4A, B*). Previously in the mouse visual cortex, similar anatomical measures of hierarchy have been performed using anterograde tracers (*D'Souza et al., 2022*) and modified retrograde rabies virus (*Yao et al., 2023*) to identify either target lamina or presynaptic cell populations. Using retrograde AAV-EF1a-H2B-EGFP and by detecting EGFP expression in the nuclei of projection neurons we find a very similar hierarchical ranking of the higher cortical visual areas as previously shown (*Figure 4—figure supplement 1*, *D'Souza et al., 2022*; *Siegle et al., 2021*; *Yao et al., 2023*).

Independent of the target area, the average fILN for both hemispheres indicates that cortical input onto VISp, SSp-bfd, and MOp reflects predominantly a feedback organization (*Figure 4C*). However, on average, and compared to the ipsilateral projection, the average fILN of the contralateral hemisphere is larger across all injection targets (*Figure 4C*, p < 0.05, one-sided paired *t*-test, respectively). Moreover, the overall contralateral input appears more narrowly distributed in its degree of anatomical hierarchy, maintaining relatively high fILN values across all brain areas (*Figure 4C, D*; fILN range ipsilateral: 0.42 ± 0.1 to 1.0 ± 0.0 vs. contralateral: 0.66 ± 0.07 to 1 ± 0). As a consequence, regardless of the target area, we found a significant negative correlation between the areal fILN values on the ipsilateral side and the magnitude of the difference in the fILN of their contralateral counterpart (*Figure 4E*, VISp: *r* = –0.71 ± 0.07, SSp-bfd: *r* = –0.7 ± 0.04, MOp: *r* = –0.77 ± 0.04, p < 0.05, Spearman correlation) although the ranking of contralateral hierarchy remained similar to that of the ipsilateral hierarchy (*Figure 4F*, ipsi- vs. contralateral fILN: VISp: *r* = 0.74 ± 0.03, SSp-bfd: *r* = 0.63 ± 0.08, MOp: *r* = 0.63 ± 0.03, p < 0.05, Spearman correlation). In line with the classical view of reciprocal feedforward/feedback connectivity (*Angelucci and Petreanu, 2023*; *Markov et al., 2014*; *Young et al., 2021*) and in the case where we can directly compare the interconnectivity of two target sources within the ipsilateral hemisphere (VISp and SSp-bfd), we see feedforward input from VISp to SSp-bfd and feedback input from SSp-bfd to VISp (*Figure 4E*, *Figure 4—figure supplement 2A, B*, magenta and green arrows). However, all contralateral areas were found to provide predominantly feedback input (*Figure 4D–G*, *Figure 4—figure supplement 1A–C*). These data also reveal unique instances, for example in SSp-bfd, where the ipsi- and contralateral input from a subset of visual areas

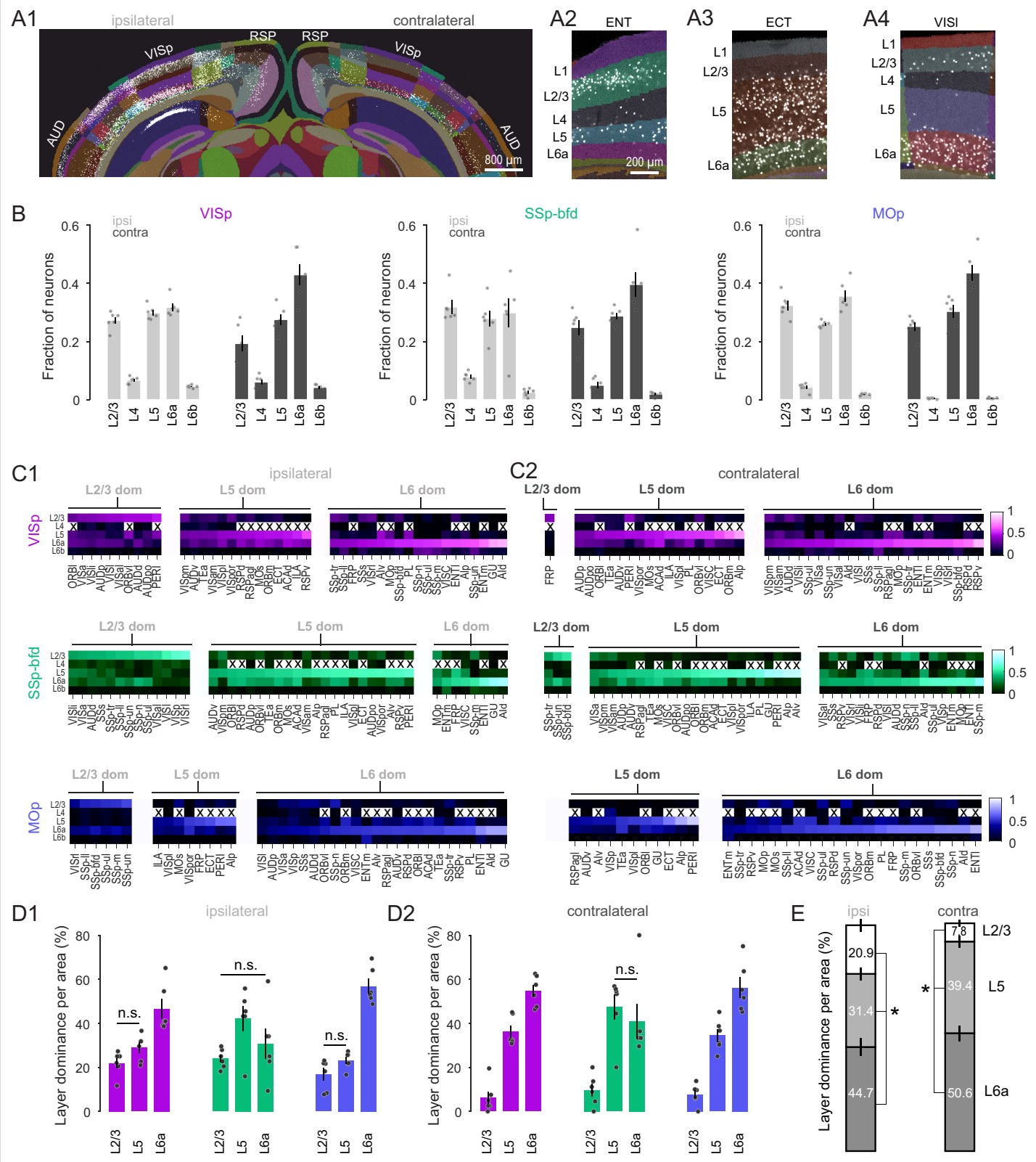

**Figure 3.** Cortical Layer 6 is a major source of input to VISp, SSp-bfd, and MOp. (**A1**) A segmented image of the cortex used to assign the laminar position of detected neurons after injection into VISp. Image of segmented entorhinal (ENT), ectorhinal (ECT), and lateral visual area (VISl) on the contralateral hemisphere showing local dominance by L2/3 (**A2**), L5 (**A3**), and L6a (**A4**), respectively. (**B**) Bar plot showing average fraction of cells (± SEM, n = 6 mice) in L2/3, L4, L5, L6a, and L6b across the ipsi- and contralateral hemisphere for the three injection targets. (**C**) Average heatmaps (n =

*Figure 3 continued on next page*

*Figure 3 continued*

6 mice) based on the fractional count across laminae for each cortical area separated according to injection target and local dominance for ipsilateral (**C1**) and contralateral (**C2**) hemispheres. Dominance quantification is based on average laminar fractional count across animals. Crosses indicate cortical areas that do not contain L4. (**D**) Bar graphs showing the average percentage of areas (± SEM, *n* = 6 mice) in which L2/3, L5, and L6 dominate the fractional cell count for ipsilateral (**D1**) and contralateral (**D2**) hemispheres. Unless indicated, all comparisons reach statistical significance (n.s. = non-significant). Dominance quantification is based on laminar fractional count per animal. (**E**) Stacked bar plots showing average laminar dominance for all cortical areas (± SEM) pooled across all injection targets. Asterisks indicate statistical significance (p < 0.05).

The online version of this article includes the following source data for figure 3:

**Source data 1.** Data corresponding to panels B, C, D and E.

showed opposing signatures in their anatomical organization. Ipsilateral mediating feedforward while contralateral providing feedback input (highlighted in red in *Figure 4E*, *Figure 4—figure supplement 1A–C*). Together, this hierarchical analysis shows that input onto these primary sensory and motor areas is mediated largely by a high proportion of L5 and L6 neurons and that both ipsi- and contralateral input is predominantly feedback (*Figure 4G*). However, in some instances ipsilateral input can appear strongly feedforward and in stark contrast to the feedback organization of projections from the same area in the contralateral hemisphere.

## Increased contralateral hierarchy is due to input from sensory and motor areas

Within the ipsilateral hemisphere, anterogradely labeled sensory–motor module projections have been shown to exhibit low hierarchy compared to lateral, medial and prefrontal modules (*Harris et al., 2019*). Using retrograde tracing and our fILN measure we find this to be the case not only for the ipsilateral hemisphere but also for the contralateral hemisphere (*Figure 5A*). Strikingly, ranking cortical modules according to their fILN values revealed extremely similarly high levels of feedback from prefrontal, medial, and lateral modules (pf-m-l) from both hemispheres. In contrast, the fILN values for visual, auditory, and somatomotor (v-a-sm) modules were different across the two hemispheres (*Figure 5A, B*) with the contralateral hemisphere providing much stronger feedback (*Figure 5A, B*, ipsilateral fILN 0.65 ± 0.02 vs. contralateral fILN 0.79 ± 0.01, p < 0.001, two-sided paired *t*-test). The difference between the ipsi- and contralateral fILN was also significantly larger for the v-a-sm compared to the pf-m-l modules when the data were partitioned according to the target areas (*Figure 5C*, p < 0.001, one-sided paired *t*-test). This therefore appears to be another generalizable principle concerning the organizational hierarchy onto the primary sensory and motor cortices. Moreover, these results suggest that the global differences in the fILN between the two hemispheres observed above are mainly due to hemispheric differences in the fILN between the v-a-sm modules.

## L6 dominates the projection from many of the intra- and interhemispheric cortical areas

We next sought to determine which cortical layers within the v-a-sm modules might account for the comparatively high contralateral fILN values. There are at least three possible scenarios that could give rise to high fILN values. Theoretically, increased fILN can stem from a reduction in the proportion of projection neurons located within L2/3, an increase in the proportion of projection neurons within L5/6, or a synergistic effect involving opposing changes in supra- and infragranular layers (*Figure 6A*). By pooling all areas (*n* = 24) within the three key sensory and motor modules for the contralateral side we observe a significant reduction in L2/3 neurons for all target areas (*Figure 6B*, ipsi- vs. contralateral L2/3 fractional counts: VISp 0.25 ± 0.02 vs. 0.15 ± 0.02, SSp-bfd 0.36 ± 0.02 vs. 0.22 ± 0.03, MOp 0.23 ± 0.01 vs. 0.1±0.01, p < 0.001, one-sided paired *t*-test) and a concomitant increase in L6 cells in VISp and SSp-bfd (*Figure 6B*, ipsilateral vs. contralateral L6 fractional counts: VISp 0.38 ± 0.02 vs. 0.42 ± 0.01, SSp-bfd 0.24 ± 0.04 vs. 0.4 ± 0.01, p < 0.05, one-sided paired *t*-test). We found there to be no change in the fraction of L5 neurons between the ipsi- and contralateral hemispheres (*Figure 6B*, ipsi- vs. contralateral L5 fractional counts: VISp 0.19 ± 0.02 vs. 0.21 ± 0.01, p = 0.38; SSp-bfd 0.27 ± 0.04 vs. 0.28 ± 0.01, p = 0.86; MOp 0.13 ± 0.02 vs. 0.12 ± 0.2, p = 0.24, two-sided paired *t*-test).

We next compared the fraction of projection neurons in a given layer to its counterpart area and layer to identify which areal subpopulations of cells might be responsible for the observed global

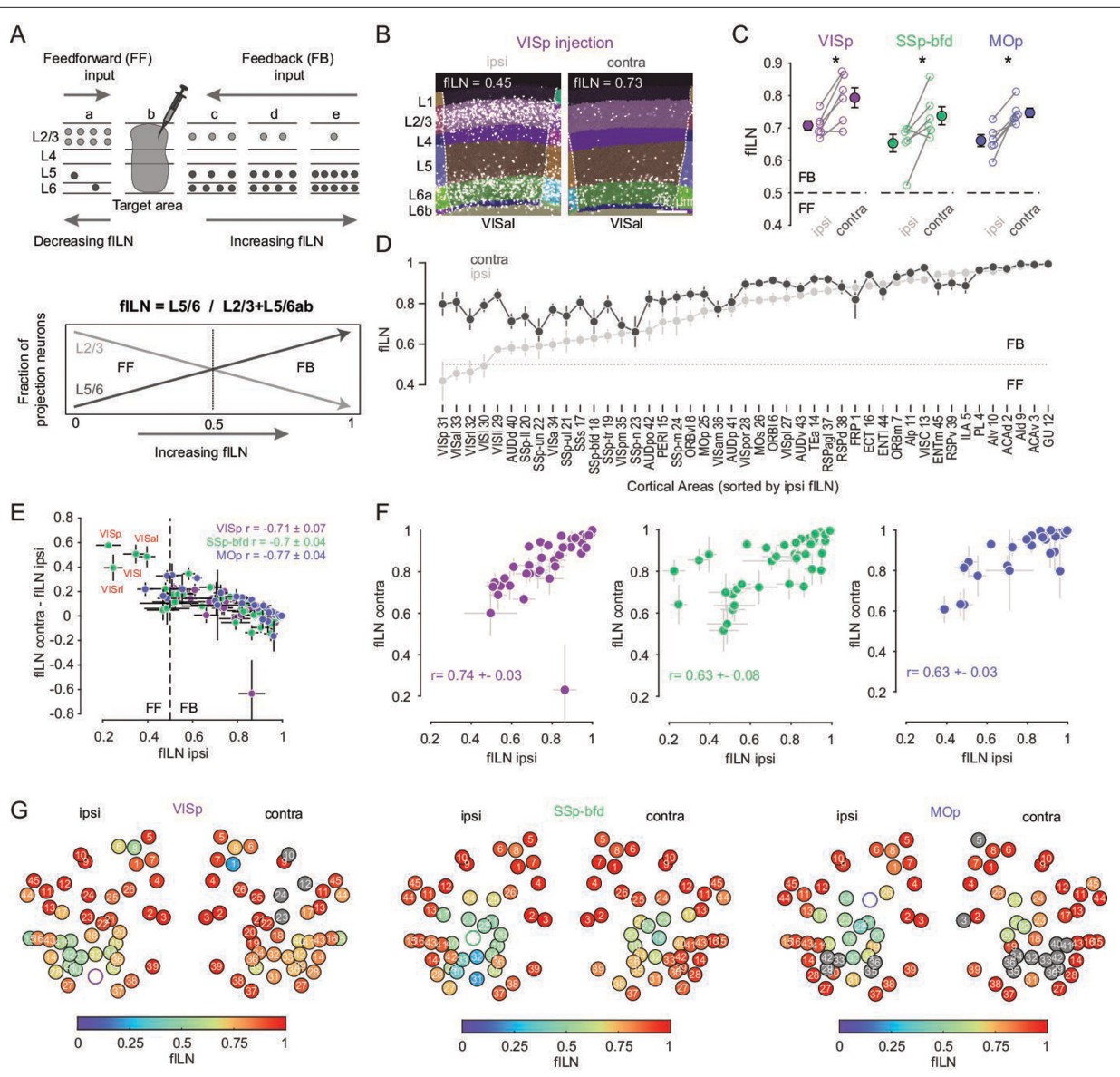

**Figure 4.** Cortical hierarchy onto VISp, SSp-bfd, and MOp. (**A**) Cartoon showing the laminar basis of the anatomical determination of cortical hierarchy (top). As the fraction of supragranular (L2/3) neurons decrease and/or the number of L5 and L6 neurons increase, the fraction of infragranular neurons (fILN) increases reflecting a predominance of feedback input (bottom). (**B**) Segmented image of the ipsi- and contralateral anterolateral visual area (VISal) projecting onto VISp showing the laminar position of detected cells. (**C**) Plots showing fILN values for all neurons within each hemisphere (blind to individual cortical areas) for each mouse for the three target injections. Average fILN values (± SEM) are depicted by filled circles. Asterisk indicates statistical significance (p < 0.05). (**D**) Plot showing the distribution of the average of ipsilateral fILN values (± SEM) for n = 45 areas (pooled across different injection experiments) and ranked from lowest to highest (gray dots). The cortical areas abbreviations and their identification numbers are displayed at the bottom. Note: the number of data points contributing to each mean value ranges from 10 to 18 depending on the number of NaN values for a given area and whether it was excluded due to it being the target injection site. The average fILN values for the same but contralateral areas are also plotted according to their corresponding ipsilateral rank (black dots). (**E**) Scatter plot showing the correlation between the ipsilateral fILN values and the difference between the contra- and ipsilateral fILN values for the three target injections. Average correlation values (± SEM, n = 6 mice) for each target area are displayed (top right). Highlighted in red are the primary (VISp), rostrolateral (VISrl), lateral (VISl), and anterolateral (VISal) visual areas projecting to SSp-bfd that display low ipsilateral fILN and high contralateral fILN values. (**F**) Scatter plots showing the correlation between the ipsi- and contralateral fILN values for the three target injections. Average correlation values (± SEM, n = 6 mice) for each target area are displayed. (**G**) Heatmap plot displaying the ipsi- and contralateral fILN values for each cortical brain area for the three injection targets. Position of the circles correspond approximately to area location. Open circles denote injection areas. Gray circles indicate cortical areas where the number of cells projecting to VISp, SSp-bfd, or MOp did not meet criterion. Numbers 1–45 indicate different brain areas, see also D.

The online version of this article includes the following source data and figure supplement(s) for figure 4:

*Figure 4 continued on next page*

**Source data 1.** Data corresponding to panels C, D, E, F, and G.

**Figure supplement 1.** Cortical hierarchy of visual areas based on fILN.

**Figure supplement 2.** Cortical hierarchy per projection area onto VISp, SSp-bfd, and MOp.

differences in the fILN (*Figure 6C*). We find there to be many areas where the fraction of L2/3 cells is significantly decreased compared to L5 and L6. L5 was more heterogeneous containing instances of both decreases and increases in the average fraction of labeled cells between the two hemispheres (*Figure 6C*). In the vast majority of cases L6 showed a relative increase in its proportion of labelled cells compared to both L2/3 and L5. Taking only those areas where we observed a significant change in the fraction of L2/3, L5, or L6 cells we found that, depending on the target area, 30–60% of sensory-motor areas showed a dramatic reduction in the fraction of contralaterally labeled L2/3 neurons compared to their ipsilateral areal counterparts. In contrast, between 30% and 50% of areas showed increases in contralateral L6 cells. On the other hand, L5 showed on average little change since its fractional contribution both increased and decreased depending on the target area (*Figure 6D*). These data show that increased fILN in the contralateral hemisphere is due to an increase in relative abundance of L6 CCs and indicates their key role in routing interhemispheric feedback.

Finally, to determine the importance of L6 in the establishment of both the ipsi- and contralateral hierarchies and the difference between the two (*Figure 4D*), we excluded L6 from the fILN calculation (*Figure 4A*). First, by comparing the default ipsilateral fILN values for all cortical areas located within the v-a-sm modules to those when L5 or L6 was excluded, we observe significant changes in fILN (*Figure 6E*, p < 0.001, two-sided paired *t*-test, Bonferroni multiple comparison correction). Importantly, the reduction in ipsi fILN was greatest when L6 was excluded compared to L5 (*Figure 6E*, p < 0.05, one-sided paired *t*-test, Bonferroni multiple comparison correction). This was also observed when performing the same comparisons on the default contralateral fILN (*Figure 6F*). Excluding L6 from the contralateral default network actually had such a significant impact that the resultant fILN was substantially lower than the default ipsilateral side (*Figure 6F*, default fILN ipsilateral 0.65 ± 0.03 vs. fILN contra excluding L6 0.52 ± 0.05, p < 0.05, one-sided paired *t*-test, Bonferroni multiple comparison correction). These data show that L6 exerts a major influence on the anatomical organization of cortical input onto VISp, SSp-bfd, and MOp and accounts for the differences in the feedback hierarchy observed within and between the two hemispheres.

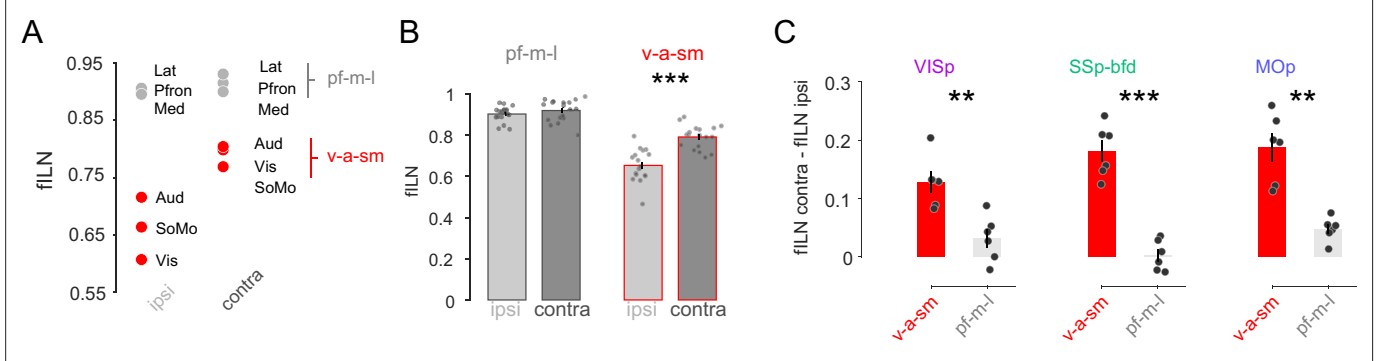

**Figure 5.** Sensory and motor but not lateral, prefrontal, and medial projections explain the hemispheric differences in cortical hierarchy. (**A**) Plot showing the average fILN values for all areas in a given module for *n* = 18 mice irrespective of target area. Red circles indicate the sensory and motor modules (visual, auditory, and somatomotor; v-a-sm) while gray circles indicate prefrontal, medial, and lateral (pf-m-l) modules. (**B**) Bar graphs comparing cross hemispheric fILN values (mean ± SEM) for prefrontal, medial, and lateral (pf-m-l) and sensory and motor (v-a-sm) modules. Each circle represents a single animal (*n* = 18). (**C**) Bar graphs (mean ± SEM) showing the interhemispheric difference in fILN values between the two modular groupings for each target projectome. Each circle represents a single animal (*n* = 6, respectively). Asterisks indicate statistical significance (** p < 0.01, *** p<0.001).

The online version of this article includes the following source data for figure 5:

**Source data 1.** Data corresponding to panels A, B and C.

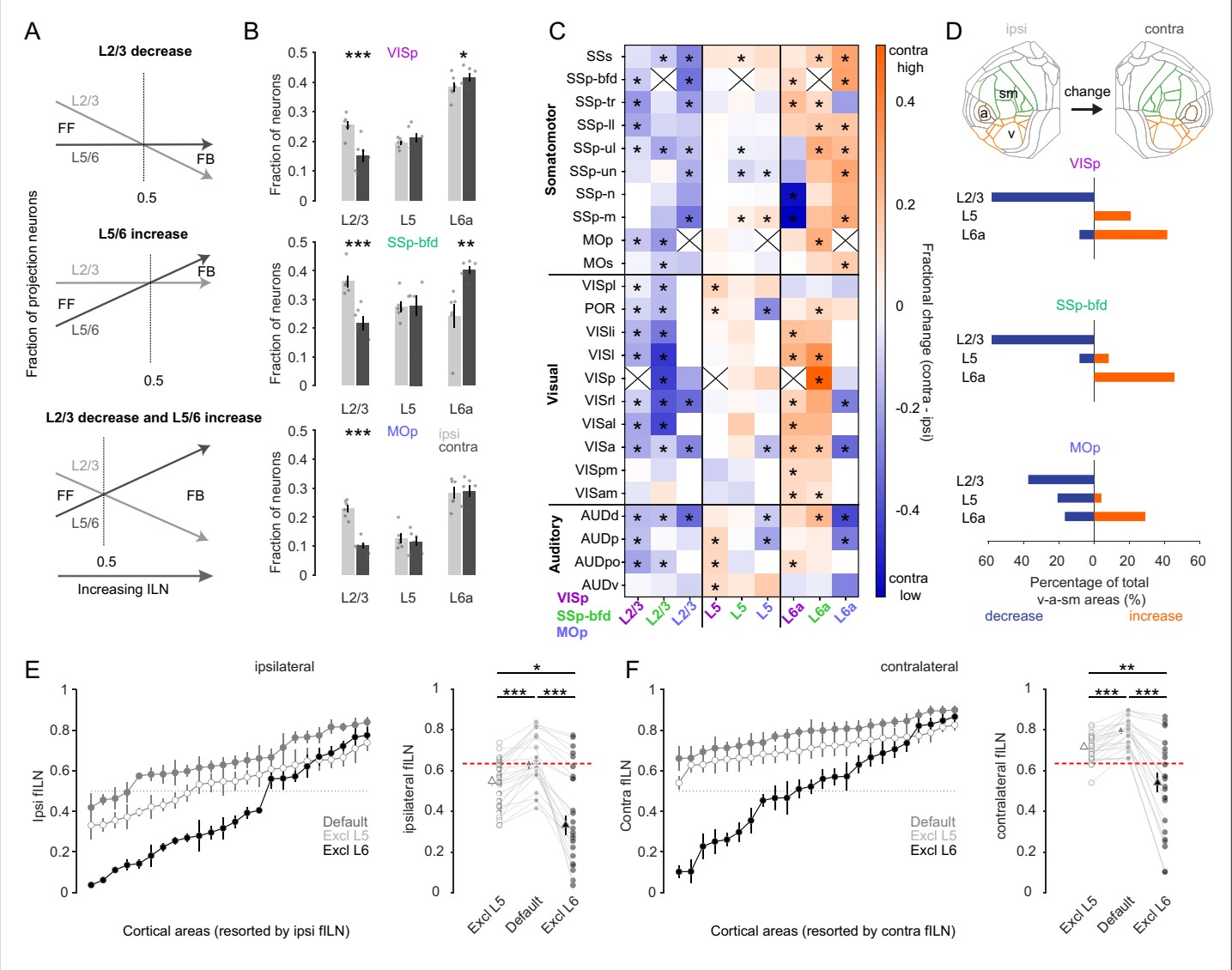

**Figure 6.** L6 dominates cortical hierarchy in the majority of sensory and motor areas. (**A**) Schematic illustrating at least three hypothetical ways in which changes in the relative distribution of L2/3, L5, and L6 neurons can give rise to increases in the fILN and reflect a predominantly feedback network. (**B**) Bar graphs showing the average (± SEM) normalized fractional count (excluding L4 and 6b) of cells for the sensory and motor modules projecting to each target area from both hemispheres (ipsi, gray; contra, black), (**C**) Heatmap displaying the fractional change of projections for L2/3, L5, and L6a between the contra- and ipsilateral hemisphere. Displayed are the 24 individual brain areas in the sensory and motor modules for the three target areas VISp, SSp-bfd, and MOp. Asterisks denote areas that significantly change between the ipsi- and the contralateral hemispheres. Crosses indicate excluded homotopic areas. (**D**) Flat map indicating the three sensory and motor modules (top). Stacked horizontal bar plots displaying the percentage of the total number cortical areas in the sensory and motor modules that significantly decrease (blue) or increase (orange; indicated by the asterisks in **C**) for the three target areas. (**E**) (Left) Plots showing the average fILN (± SEM) for each sensory and motor area (pooled across all target areas, *n* = 18 mice) and ranked according to the ipsilateral values for the default network (gray filled circles), the default network excluding L5 (open circles), and the default network excluding L6 (black filled circles). (Right) Scatter plots showing the average ipsilateral fILN scores for each of the 24 areas of the default network, the default network excluding L5 and the default network excluding L6. Filled triangles represent the mean (± SEM) of the average fILN values. Red line indicates the mean of the ipsi default network. (**F**) (Left) Plots showing the average fILN (± SEM) for each sensory and motor area (pooled across all target areas, *n* = 18 mice) and ranked according to the ipsilateral values for the default network (gray filled triangles), the contralateral values for the default network (gray filled circles), the contralateral values for the default network excluding L5 (open circles), and the default network excluding L6 (black filled circles). (Right) Scatter plots showing the average fILN scores for each of the 24 areas of the ipsi and contra default network, the default contra network excluding L5 and the default contra network excluding L6. Filled triangles represent the mean (± SEM) of the average fILN values. Red line indicates the mean of the ipsi default network. Asterisks indicate statistical significance (* p<0.05, ** p < 0.01, *** p<0.001).

The online version of this article includes the following source data for figure 6:

**Source data 1.** Data corresponding to panels B, C, D, E and F.

## Discussion

By performing comprehensive bilateral cortical circuit mapping onto VISp, SSp-bfd, and MOp we find that both primary sensory and motor areas receive extensive input from the majority of areas not only located in the ipsilateral (*Ährlund-Richter et al., 2019*; *Brown et al., 2021*; *Hafner et al., 2019*; *Muñoz-Castañeda et al., 2021*; *Sun et al., 2019*; *Yao et al., 2023*) but also contralateral cortical hemisphere (*Goulas et al., 2017*; *Yao et al., 2023*). This indicates significant cross-modal interactions not only within but also between the two cortical hemispheres. With only a few exceptions, we find that the identity, hierarchical rank and relative projection weights of input areas is highly conserved across the two hemispheres. One such exception relates to the contralateral lateral module whose composite cortical areas, compared to the ipsilateral module, display a stronger connection onto VISp (*Yao et al., 2023*), SSp-bfd and MOp indicating it may be a unique but generalizable feature of interhemispheric connectivity.

Earlier studies indicate that input from the other hemisphere onto a given area arises primarily and almost exclusively from the same contralateral area (*Fenlon and Richards, 2015*; *Yorke and Caviness, 1975*; *Zhou et al., 2013*). While we find many labeled cells in the contralateral target area, our data indicate that heterotopic connections are a prominent anatomical feature of the interhemispheric network (*Goulas et al., 2017*; *Swanson et al., 2017*; *Szczupak et al., 2023*; *Yao et al., 2023*). For both primary sensory areas the majority of the contralateral input comes from heterotopic areas located within target areas' home module followed by areas located within most of the remaining modules. In contrast, MOp appears to receive very little input from areas relating to medial, visual, and auditory modules.

Neurons projecting from the contralateral hemisphere are believed to primarily reside within L2/3 and to a lesser extent in L5 (*Fame et al., 2011*; *Pal et al., 2024*; *Ramos et al., 2008*; *Yorke and Caviness, 1975*). From an anatomical organization perspective, this would be designated as predominantly feedforward (*Markov et al., 2014*; *Markov et al., 2013a*; *Markov and Kennedy, 2013b*). We however find that L2/3 dominates the projection from the least number of cortical areas. Rather for the majority of areas, L6 emerges as a major source of both the intra- and especially the interhemispheric projection (*Liang et al., 2021*; *Olavarria and Van Sluyters, 1983*; *Yao et al., 2023*), especially for VISp and MOp. The combined dominance of L5 and L6 projecting neurons shown here indicates that the vast majority of cortical projections to primary sensory and motor regions are feedback in nature.

One explanation regarding the apparent discrepancy in these two sets of observations may be due to technical differences and limitations of different tracing approaches. Chemical or protein-based anterograde and retrograde tracers such as Cholera toxin subunit B (CTB) have been widely used but have low transduction efficiency (*Saleeba et al., 2019*; *Weiler et al., 2024*) compared to recently developed retrograde viruses (*Tervo et al., 2016*). Second, mouse cre lines are a common tool for targeting specific cells in specific layers (*Bortone et al., 2014*; *Harris et al., 2019*; *Yao et al., 2023*). However, as is the case for L6 CCs (*Vélez-Fort et al., 2014*), not all cell types are accessible using cre-driver lines. In our study, we targeted all cell types located throughout all laminae within the target area that is in contrast to previous tracing studies that appear biased toward upper layers of the injection area (*Adaikkan et al., 2022*; *Chovsepian et al., 2017*; *Massé et al., 2017*). It also seems unlikely that the observed predominance of L6 labeling revealed here can be attributed to labeling bias since our viral injections effectively covered all cortical layers and excluded the white matter. Moreover, multiple contra- and ipsilateral areas exhibit significant labeling in L2/3 and/or 5 and prior studies using non-viral tracers such as CTB or Fluorogold have similarly reported a dominance of L6 projection neurons in the same specific pathways (e.g. SSp-bfd → VISp, *Bieler et al., 2017*; *Weiler et al., 2024*) as those revealed here using a retro-AAV-based approach.

The idea that cortical feedforward and feedback circuits may have an anatomical signature has been extensively interrogated in the mammalian visual cortical system (*D'Souza et al., 2022*; *Felleman and Van Essen, 1991*; *Harris et al., 2019*; *Markov et al., 2014*; *Yao et al., 2023*) where there exists a plethora of physiological data indicating functional hierarchy. In primates, feedforward and feedback neurons tend to reside in different cortical layers (*Markov et al., 2014*) and the ratio of supra- to infra-granular neurons has been employed to anatomically define visual cortical hierarchy (*Barone et al., 2000*; *Markov et al., 2014*).

Here, we demonstrate in mice that the fILN metric effectively ranks higher visual areas that is consistent with the anatomical organization derived from axon termination patterns, as well as retrograde

rabies tracing (*D'Souza et al., 2022*; *Harris et al., 2019*; *Yao et al., 2023*) and physiological experiments that describe functional hierarchy in mice (*D'Souza et al., 2022*; *Jia et al., 2022*; *Siegle et al., 2021*). By applying fILN-based area ranking cortex-wide, we uncover a global organization that is largely consistent with the hierarchical ranking of cortical areas based on anterograde labeling (*Harris et al., 2019*), placing sensory–motor modules at the bottom and lateral, medial, and prefrontal modules at the top of cortical hierarchy. Our data therefore demonstrate that the fILN metric can serve as a continuous parameter whose ranking may be used as an estimate of feedforward and feedback hierarchy for understanding cortical connectivity.

Typically, feedforward projections are characterized as 'drivers' while feedback projections are seen as 'modulators' of neuronal activity (*Markov et al., 2013a*). Our anatomical data reveal that VISp, SSp-bfd, and MOp mostly receive feedback from the ipsilateral cortex. This is generally expected as these targets are the first cortical areas to receive sensory input or to initiate motor functions. The predominance of feedback projections suggests a modulatory role rather than driving input onto these primary cortical areas. From our data, the pairwise intra-hemispheric connectivity between VISp and SSp-bfd aligns well with historical theories of cortical hierarchy, which suggest that feedforward connections are typically complemented by reciprocal feedback connections, at least within the ipsilateral hemisphere (*Angelucci and Petreanu, 2023*; *Felleman and Van Essen, 1991*; *Markov et al., 2013a*; *Young et al., 2021*). Our data also reveal a notable exception that appears to involve ipsilateral primary and higher visual cortical areas providing cross-modal feedforward input onto SSp-bfd. On the other hand, input from the same areas on the contralateral side are feedback in their organization as one might expect to observe for higher-order areas. The fact that, regardless of the target area, contralateral projections are predominantly feedback suggests that the conventional rule of reciprocally feedforward and feedback projections does not apply for interhemispheric input at least to primary areas VISp, SSp-bfd, or MOp. It will nevertheless be interesting and important to further investigate the hierarchical organization of cortical input onto higher cortical areas such anterior cingulate and retrosplenial cortex or within subregions of a given sensory area (e.g. higher compared to lower visual cortical areas).

Due to their dominant feedback bias, contralateral inputs may act as modulators of cortical activity within their ipsilateral target regions (*Innocenti et al., 2022*; *Markov et al., 2014*). For instance, inactivation of contralateral somatosensory areas in monkeys increases the receptive field size in the ipsilateral somatosensory area, suggesting a loss of modulatory contralateral input (*Clarey et al., 1996*). Moreover, contralateral auditory projections act to sharpen frequency tuning in its ipsilateral counterpart (*Slater and Isaacson, 2020*) and activation of contralateral visual projections strongly modulates visual cortical responses in the binocular zone of VISp (*Zhao et al., 2013*). However, these studies focused purely on homotopic areas while the functions of heterotopic areas largely remain elusive. Given the observed extent of heterotopic input, future studies should address their functional roles in hemispheric communication and ultimately in behavior.

By showing that hierarchical disparities between the two hemispheres are largely accounted for by differences in the input from the sensory–motor modules, we offer a more nuanced understanding of interhemispheric communication and a prominent role for L6 that is strategically positioned within the cortex. Additionally, we show that increased fILN is predominantly due to the target areas receiving input from L6 CCs rather than L6 corticothalamic cells. L6 CCs not only integrate input from and output to its local cortical column (*Vélez-Fort et al., 2014*) but also receive substantial thalamic input, at least in sensory areas (*Crandall et al., 2017*). Moreover, responses of L6 CCs to external stimuli temporally precedes those in other cortical layers (*Egger et al., 2020*). Thus, L6 CCs are ideal candidates to rapidly relay information to other cortical areas thereby providing feedback modulation (*Weiler et al., 2024*), suggestive of predictive coding (*Rao and Ballard, 1999*), or the transmission of efference copy (*Latash, 2021*; *Vallortigara, 2021*; *von Holst and Mittelstaedt, 1950*).

## Materials and methods
### Animals
All experiments were performed on 6- to 23-week-old mice. For the data presented in *Figures 1–6*, 18 mice were used from which 11 were males and 7 were females. From these 18 mice, 14 were wild-type C57BL/6J mice, 3 were Gad2tm2(cre)Zjh/J (GAD-cre), and 1 mouse was B6.FVB(Cg)-Tg(Ntsr1-cre)

Gn220Gsat/Mmucd (Ntsr1-cre). The GAD-cre and Ntsr1-cre mice were crossed with B6.Cg-GT(ROSA)26 Sortm14(CAG-tdTomato)Hze/J (Ai14, Cre-dependent tdTomato reporter) to achieve TdTomato reporter mice. Additionally, three Ntsr1-cre × Ai14 and three GAD-cre × Ai14 mice were used for the data presented in *Figure 1—figure supplement 4*. Mice were raised in standard cages on a 12-hr light/dark cycle, with food and water available ad libitum. In this study, six VISp-injected mice were previously used to quantify areal input from the ipsilateral hemisphere (*Weiler et al., 2024*).

All surgeries and experiments were conducted in accordance with the UK Home Office regulations (Animal (Scientific Procedures) Act 1986), approved by the Animal Welfare and Ethical Review Body (AWERB; Sainsbury Wellcome Centre for Neural Circuits and Behavior) and in compliance with ARRIVE guidelines. Every effort was made to minimize the number of animals and their suffering.

## Surgical procedures and viral injections

All surgical procedures were carried out under isoflurane (2–5%) and after carprofen (5 mg/kg, s.c.) had been administered. For retrograde viral tracing, we used rAAV2-retro-EF1a-H2B-EGFP (Nuclear retro-AAV, titer: $8.8 \times 10^{13}$ GC per ml). Mice were anesthetized under isoflurane (~2%) and craniotomies performed. Virus injection was performed using borosilicate glass injection pipettes (Wiretrol II; Drummond Scientific) pulled to a taper length of ~30 mm and a tip diameter of ~50 µm. Virus was delivered at a rate of 1–2 nl/s using Nanoject III (Drummond Scientific, USA) and injected at three cortical depths covering all layers of the VISp, SSp-bfd, and MOp, respectively (*Figure 1—figure supplement 3*). After injections, the craniotomy was sealed with silicon (kwik-cast), the skin was re-sutured and animals were allowed to recover for 2–4 weeks. Injection coordinates for the monocular and binocular zone of VISp, SSp-bfd, and MOp were based on the Allen Reference Atlas (coronal, 2D, *Wang et al., 2020*).

## Perfusion and brain extraction

For perfusions, mice were first deeply anesthetized using Pentobarbital Sodium (10 mg/kg). A blunt needle was placed in the left ventricle, whilst an incision was performed in the right atrium of the heart. Following this, blood was first cleared using 100 mM PBS. Subsequently, the animal was perfused with saline containing 4% PFA. After successful fixation, the head was removed and the brain dissected out. The brain was further fixed in 4% PFA overnight at 4°C, and then stored in 100 mM PBS at 4°C until ready for imaging.

## Brain wide serial two-photon imaging

For serial section two-photon imaging, on the day of imaging, brains were removed from the PBS and dried. Brains were then embedded in agarose (4%) using a custom alignment mold to ensure that the brain was perpendicular to the imaging axis. The agarose block containing the brains were trimmed and then mounted onto the serial two-photon microscope containing an integrated vibrating microtome and motorized *x–y–z* stage (STP tomography, *Osten and Margrie, 2013*; *Ragan et al., 2012*). For this, a custom system controlled by ScanImage (v5.6, Vidrio Technologies, USA) using BakingTray (https://bakingtray.mouse.vision/) was used. Imaging was performed using 920 nm illumination. Images were acquired with a 2.3 × 2.3 µm pixel size, and 5 µm plane spacing. Eight to ten optical planes were acquired over a depth of 50 µm in total. To image the entire brain, images were acquired as tiles and then stitched using StitchIt (https://doi.org/10.5281/zenodo.3941901). After each mosaic tile was imaged at all optical planes, the microtome automatically cut a 50-µm slice, enabling imaging of the subsequent portions of the sample and resulting in full 3D imaging of entire brains. All images were saved as a series of 2D TIFF files.

Images were registered to the Allen Mouse Brain Common Coordinate Framework (*Wang et al., 2020*) using the software brainreg (*Tyson et al., 2022*) based on the aMAP algorithm (*Niedworok et al., 2016*). All atlas data were provided by the BrainGlobe Atlas API (*Claudi et al., 2020*). For registration, the sample image data were initially down-sampled to the voxel spacing of the atlas used and reoriented to align with the atlas orientation using bg-space (https://doi.org/10.5281/zenodo.4552537). The 10 µm atlas was used for cell detection and mapping. To manually segment viral injection sites, the software brainreg-segment (*Tyson et al., 2022*) was used. Automated cell detection and deep learning based cell classification was performed using the cellfinder software (*Tyson et al.,*

2021) and cross-validated with manual annotation (see validation in *Weiler et al., 2024*). All analysis in this manuscript was performed in atlas space (*Wang et al., 2020*).

Figures showing detected cells in 3D atlas space were generated using the brainrender software (*Claudi et al., 2021*) and custom scripts written in Python 3.9.

## Confocal imaging and analysis

For a subset of Ntsr1-cre × Ai14 and GAD-cre × Ai14, slices cut with STP topography were kept, post-fixed and mounted for confocal imaging (Leica SP8). Individual slices were imaged using tile-scan acquisitions with a 10× air objective. Voxel sizes were 2 μm (*x/y*) and 3–6 μm (*z*). We performed sequential imaging of GFP and td-tomato signals to achieve optimal spectral separation. We counted the overlap between GAD+ or NSTR1+ and retro-GFP+ cells using the Cell Counter feature of Fiji. Within the dataset only approximately 1% of retro-GFP+ cells were GAD+ (*Figure 1—figure supplement 4*) with no overlap between retro-GFP+ and NSTR1+ cells (*Figure 1—figure supplement 4*).

## Data analysis

Cellfinder outputs a CSV file containing the laminar count of detected nuclei for each cortical area. This was analyzed using custom-written code in MATLAB 2023–24 and Python 3.9. We applied the following criteria for data exclusion: (1) Instances where more than 30% of the viral injection bolus was located outside the respective target area (VISp, SSp-bfd, or MOp, *Figure 1—figure supplement 2A–D*). (2) A given cortical area in a given hemisphere had to contain 10 or more cells. In the case where a given area contained less than 10 cells the area cell was set at zero. (3) Additionally to be included in the analysis a given area had to contain more than 10 cells in at least three injections and in the same hemisphere of three different mice. For comparison of the laminar distribution of cells within different brain areas, values were normalized to the total number of cells detected in each area. The fILN was calculated as following:

$$fILN = \left( L5_{fractionarea} + L6_{fractionarea} \right) / \left( L2/3_{fractionarea} + L5_{fractionarea} + L6_{fractionarea} \right).$$

.

## Statistics

Details of all *n* numbers and statistical analysis are provided either in the results and/or in the figure captions. Before comparison of data, individual datasets were checked for normality using the Anderson–Darling test in MATLAB 2023–24. The required sample sizes were estimated based on literature and our past experience performing similar experiments (*Brown et al., 2021*; *Weiler et al., 2024*). Significance level was typically set as p < 0.05 if not stated otherwise. Statistical analyses were performed using MATLAB 2023–24. Asterisks indicate significance values as follows: *p < 0.05, **p < 0.01, ***p < 0.001.

## Acknowledgements

We thank Panagiota Iordanidou for excellent technical support and assistance as well as Mateo Vélez-Fort for comments on the manuscript. The authors are further grateful to the support staff of the Neurobiological Research Facility at Sainsbury Wellcome Centre. The authors would like to thank the SWC Advanced Microscopy Facility for their assistance with two-photon tomography and confocal imaging. TWM and SW are funded by The Wellcome Trust (214333/Z/18/Z; 219627/Z/19/Z; 306384/Z/23/Z), Gatsby Charitable Foundation (GAT3361) and Humboldt Foundation (SW). MT is funded by the Interdisciplinary Centre for Clinical Research (IZKF; Advance medical scientist – Program 11).

# Additional information

## Funding

| Funder | Grant reference number | Author |
|---|---|---|
| Wellcome Trust | 219627/Z/19/Z | Troy W Margrie |
| Humboldt Foundation | Feodor Lynen Research Fellowship | Simon Weiler |
| Interdisciplinary Centre for Clinical Research | Advance medical scientist - Program 11 | Manuel Teichert |
| Wellcome Trust | 10.35802/214333 | Troy W Margrie |
| Wellcome Trust | 10.35802/306384 | Troy W Margrie |
| Gatsby Charitable Foundation | GAT3361 | Troy W Margrie |

The funders had no role in study design, data collection, and interpretation, or the decision to submit the work for publication. For the purpose of Open Access, the authors have applied a CC BY public copyright license to any Author Accepted Manuscript version arising from this submission.

## Author contributions

Simon Weiler, Conceptualization, Resources, Data curation, Software, Formal analysis, Supervision, Funding acquisition, Validation, Investigation, Visualization, Methodology, Writing – original draft, Project administration, Writing – review and editing; Manuel Teichert, Conceptualization, Data curation, Software, Formal analysis, Supervision, Validation, Investigation, Visualization, Methodology, Writing – original draft, Project administration, Writing – review and editing; Troy W Margrie, Conceptualization, Resources, Formal analysis, Supervision, Funding acquisition, Validation, Visualization, Writing – original draft, Project administration, Writing – review and editing

## Author ORCIDs

Simon Weiler  https://orcid.org/0000-0003-4731-8369
Manuel Teichert  http://orcid.org/0009-0008-8129-6849
Troy W Margrie  https://orcid.org/0000-0002-5526-4578

Reviewer #1 (Public review): https://doi.org/10.7554/eLife.100478.3.sa1
Reviewer #2 (Public review): https://doi.org/10.7554/eLife.100478.3.sa2
Author response https://doi.org/10.7554/eLife.100478.3.sa3

# Additional files

## Supplementary files

MDAR checklist

## Data availability

Processed data and source data for individual figures are available at https://doi.org/10.5061/dryad.8kprr4xzd. Code to replicate the analysis and figures is available at https://github.com/simonweiler/layer6_anatomy (copy archived at *Weiler, 2025*). For access to the raw image stacks, please directly contact t.margrie@ucl.ac.uk.

The following dataset was generated:

| Author(s) | Year | Dataset title | Dataset URL | Database and Identifier |
|---|---|---|---|---|
| Weiler S | 2025 | Data structures and source data for the manuscript: "Layer 6 corticocortical neurons are a major route for intra and interhemispheric feedback" by Simon Weiler, Manuel Teichert and Troy W. Margrie" | https://doi.org/10.5061/dryad.8kprr4xzd | Dryad Digital Repository, 10.5061/dryad.8kprr4xzd |

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
