## [Editor Report · eLife Assessment]

This **important** study compares the cortical projections to primary motor and sensory areas originating from the ipsilateral and contralateral hemispheres. Results show that, while there is substantial symmetry between the two hemispheres regarding the areas sending projections to these primary cortical areas, contra-hemispheric projections had more inputs from layer 6 neurons than ipsi-projecting ones. The evidence is **compelling** and the conclusions are supported by rigorous analyses.

---

## [Referee Report · Reviewer #1 (Public review)]

Weiler, Teichert, and Margrie systematically analyzed long-range cortical connectivity, using a retrograde viral tracing strategy to identify layer and region-specific cortical projections onto the primary visual, primary somatosensory, and primary motor cortices. Their analysis revealed several hundred thousand inputs into each region, with inputs originating from almost all cortical regions but dominated in number by connections within cortical sub-networks (e.g. anatomical modules). Generally, the relative areal distribution of contralateral inputs followed the distribution of corresponding ipsilateral inputs. The largest proportion of inputs originated from layer 6a cells, and this layer 6 dominance was more pronounced for contralateral than ipsilateral inputs, which suggests that these connections provide predominantly feedback inputs. The hierarchical organization of input regions was similar between ipsi- and contralateral regions, except for within-module connections, where ipsilateral connections were much more feed-forward than contralateral. These results contrast earlier studies which suggested that contralateral inputs only come from the same region (e.g. V1 to V1) and from L2/3 neurons. The conclusions of this paper are well-supported by the data and analysis, and useful follow-up analyses and discussions are present in the supplemental figures. Taken together, these results provide valuable data supporting a view of interhemispheric connectivity in which layer 6 neurons play an important role in providing modulatory feedback.

---

## [Referee Report · Reviewer #2 (Public review)]

Summary:

Weiler et al use retrograde tracers, two-photon tomography, and automatic cell detection to provide a detailed quantitative description of the laminar and area sources of ipsi- and contralateral cortico-cortical inputs to two primary sensory areas and a primary motor area. They found considerable bilateral symmetry in the areas providing cortico-cortical inputs. However, although the same regions in both hemispheres tended to supply inputs, a larger proportion of inputs from contralateral areas originated from deeper layers (L5 and L6).

Strengths:

The study applies state-of-the-art anatomical methods, and the data is very effectively presented and carefully analyzed. The results provide many novel insights on the similarities and differences of inputs from the two hemispheres. While over the past decade there has been many studies quantitively and comprehensively describing cortico-cortical connections, by directly comparing inputs from the ipsi and contralateral hemispheres, this study fills in an important gap in the field. It should be of great utility and an important reference for future studies on inter hemispheric interactions.

Weaknesses:

Overall, I do not find any major weakness in the analyses or their interpretation. However, one must keep in mind that the study only analyses inputs projecting to three areas. This is not an inherent flaw of the study; however, it warrants caution when extrapolating the results to callosal projections terminating in other areas. As inputs to two primary sensory areas and one is the primary motor cortex are studied, some of the conclusions could potentially be different for inputs terminating in high-order sensory and motor areas. Given that primary areas were injected, there are few instances of feedforward connections sampled in the ipsilateral hemisphere. The study finds that while ipsi- projections from visual cortex to barrel cortex are feedforward given its fILN values, those from the contralateral visual cortex are feedback instead. This is now acknowledged in the revised discussion.

Another issue that is left unexplored is that, in the current analyses the barrel and primary visual cortex are analyzed as a uniform structure. It is well established that both the laminar sources of callosal inputs and their terminations differ in the monocular and binocular areas of the visual cortex (border with V2L). Similarly, callosal projections differ when terminating the border of S1 (A row of whiskers) then in other parts of S1. Thus, some of the conclusions regarding the laminar sources of callosal inputs might depend on whether one is analyzing inputs terminating or originating in these border regions. This is now acknowledged in the revised version.

---

## [Author Response]

The following is the authors’ response to the original reviews.

**Reviewer #1 (Public review):**
Weiler, Teichert, and Margrie systematically analyzed long-range cortical connectivity, using a retrograde viral tracing strategy to identify layer and region-specific cortical projections onto the primary visual, primary somatosensory, and primary motor cortices. Their analysis revealed several hundred thousand inputs into each region, with inputs originating from almost all cortical regions but dominated in number by connections within cortical sub-networks (e.g. anatomical modules). Generally, the relative areal distribution of contralateral inputs followed the distribution of corresponding ipsilateral inputs. The largest proportion of inputs originated from layer 6a cells, and this layer 6 dominance was more pronounced for contralateral than ipsilateral inputs, which suggests that these connections provide predominantly feedback inputs. The hierarchical organization of input regions was similar between ipsi- and contralateral regions, except for within-module connections, where ipsilateral connections were much more feed-forward than contralateral. These results contrast earlier studies which suggested that contralateral inputs only come from the same region (e.g. V1 to V1) and from L2/3 neurons. Thus, these results provide valuable data supporting a view of interhemispheric connectivity in which layer 6 neurons play an important role in providing modulatory feedback.The conclusions of this paper are mostly well-supported by the data and analysis, but additional consideration of possible experimental biases is needed.

We thank the reviewer for their positive feedback on our manuscript.

Further discussion or analysis is needed about possible biases in uptake efficiency for different cell types. Is it possible that the nuclear retro-AAV has a tropism for layer 6 axons? Quantitative comparisons with results obtained with alternative methods such as rabies virus (Yao et al., 2023) or anterograde tracing (Harris et al., 2019) may be helpful for this.

We appreciate this technical comment. For the reasons indicated below we are confident that our AAV approach successfully and rather comprehensively labels inputs to the three target areas. Firstly, in the brains in which we injected our retrograde nuclear-AAV tracer into VISp, SSp-bfd or MOp we found several instances where layer 5 and/or layer 2/3 as was the dominant cortical projection layer (please see e.g. Figure 3 heatmaps). This was true for both ipsilateral and contralateral projection.

Secondly, by way of comparison Yao et al., 2023 injected rabies virus into VISp (but not in SSp-bfd or MOp) and their results show notable similarities to ours: (1) They show that contralateral inputs to VISp (and higher visual areas) were mainly located in Layers 5 and 6. (2) Retrogradely labelled neurons in higher visual areas revealed anatomical hierarchy that reflects the known functional hierarchy of the mouse cortical visual system and that shown by our retro-AAV approach. Thus, as AAV and rabies based tracing lead to similar results, this is further evidence against bias via tropism of our AAV tracer. That said, direct comparisons of the results between our study and the Yao et al., 2023 study should be viewed with some caution since Yao et. al. injected rabies virus into specific Cre-driver lines in which the rabies virus targets individual genetically defined cell types in specific layers. Importantly, because of the lack of a specific cre-driver line, L6 cortico-cortical (L6 CC) cells could not be targeted by their approach. Thus, the dataset in Yao et al., overlook the contribution of L6 CCs due to the lack of available Cre-lines.

Thirdly, in a recent study (Weiler et al., 2024) we found that in a specific pathway (SSp-bfd→ VISp) both retro-AAV and the more traditional non-viral tracer cholera toxin subunit B (CTB) identified neurons in Layer 6 as the main source of projection neurons. The same results for the same pathway was shown by Bieler et al., 2019 (Bieler et al., 2017) using Fluorogold for retrograde tracing. Thus, the described dominance of Layer 6 projection neurons in specific pathways is likely not the result of a tropism of retro-AAV tracers.

Please also see that we have now further extended the summary of these points in our revised manuscript in the discussion section (e.g. lines 457-463):

Quantitative analysis of the injection sites should be included to account for possible biases. For example, L6 neurons are known to be the main target of contralateral inputs into the visual cortex (Yao et al., 2023). Thus, if the injections are biased towards or against layer 6 neurons, this may change the layer distribution of retrogradely labeled input cells. Comparison across biological replicates may help reveal sensitivity to particular characteristics of the injections.

In response to the reviewers' feedback, please see we have now quantified the injection volume per cortical layer, as shown in the revised Fig. S3D. Our results indicate that the injections were not biased toward Layer 6. Instead, the injected tracer volumes in Layers 1, 4, 5, and 6 were similar across all animals and injected areas. However, we observed that the injected tracer volume in Layer 2/3 tended to be higher than in other layers. Although the tracer volumes in Layers 2/3 appeared to be higher, the proportion of input neurons located in Layers 2/3 for most of the cortical projection areas was consistently lower than that from Layer 6. These findings provide strong evidence against injection bias towards L6 inputs.

The possibility of labelling axons of passage within the white matter should be addressed. This could potentially lead to false positive connections, contributing to the broad connectivity from most cortical regions that were observed.

For clarification, please see Fig.S2B in our revised manuscript. In this panel we plot the average percentage volume of the viral boli in the target areas and in all other nearby structures including the white matter. The percentage of virus injected into the white matter (WM) was 0.0824 ± 0.0759% for VISp and 0.0650 ± 0.0481 for SSp-bfd injections. Notably, injections into MOp showed no leakage into white matter (0%). These minimal volumes of virus in the white matter are unlikely to significantly influence the observed profile of widespread connectivity. Please see we have added a sentence to the Results section (lines 84-86) where we state that we only used brains that had a transduction of the white matter below 0.1%.

**Reviewer #2 (Public review):**
Summary:Weiler et al use retrograde tracers, two-photon tomography, and automatic cell detection to provide a detailed quantitative description of the laminar and area sources of ipsi- and contralateral cortico-cortical inputs to two primary sensory areas and a primary motor area. They found considerable bilateral symmetry in the areas providing cortico-cortical inputs. However, although the same regions in both hemispheres tended to supply inputs, a larger proportion of inputs from contralateral areas originated from deeper layers (L5 and L6).Strengths:The study applies state-of-the-art anatomical methods, and the data is very effectively presented and carefully analyzed. The results provide many novel insights into the similarities and differences of inputs from the two hemispheres. While over the past decade there have been many studies quantitatively and comprehensively describing cortico-cortical connections, by directly comparing inputs from the ipsi and contralateral hemispheres, this study fills in an important gap in the field. It should be of great utility and an important reference for future studies on inter-hemispheric interactions.

We thank the reviewer for this encouraging feedback on our manuscript.

Weaknesses:Overall, I do not find any major weakness in the analyses or their interpretation. However, one must keep in mind that the study only analyses inputs projecting to three areas. This is not an inherent flaw of the study; however, it warrants caution when extrapolating the results to callosal projections terminating in other areas. As inputs to two primary sensory areas and one is the primary motor cortex are studied, some of the conclusions could potentially be different for inputs terminating in high-order sensory and motor areas. Given that primary areas were injected, there are few instances of feedforward connections sampled in the ipsilateral hemisphere. The study finds that while ipsi-projections from the visual cortex to the barrel cortex are feedforward given its fILN values, those from the contralateral visual cortex are feedback instead. One is left to wonder whether this is due to the cross-modal nature of these particular inputs and whether the same rule (that contralateral inputs consistently exhibit feedback characteristics regardless of the hierarchical relationship of their ipsilateral counterparts with the target area,) would also apply to feedforward inputs within the same sensory cortices.

We acknowledge that what we find for primary sensory and motor target areas may not hold for other functionally different areas such as anterior cingulate cortex, retrosplenial cortex or frontal lobe that might be expected to receive strong feedforward cortical input. To begin to understand the organization of the global cortical input we have however first explored with primary sensory and motor areas. Please see that we have now added a sentence to the Discussion section of our manuscript that highlights the importance of investigating the hierarchical organization of intra and interhemispheric input onto higher cortical areas or within subregions of a given sensory area.

Another issue that is left unexplored is that, in the current analyses the barrel and primary visual cortex are analyzed as a uniform structure. It is well established that both the laminar sources of callosal inputs and their terminations differ in the monocular and binocular areas of the visual cortex (border with V2L). Similarly, callosal projections differ when terminating the border of S1 (a row of whiskers), and then in other parts of S1. Thus, some of the conclusions regarding the laminar sources of callosal inputs might depend on whether one is analyzing inputs terminating or originating in these border regions.

The aim of the present study was to analyse the global projectome to the VISp, SSp-bfd and MOp, irrespective of which subregions were included. Importantly, we purposely injected rather large bolus volumes to achieve large sample sizes of target neurons in each cortical layer. For SSp-bfd, we utilised our previously reconstructed barrel map (Weiler et al., 2024) to precisely map our viral injection sites onto the barrels (Author response image 1). Analysis revealed that the six injection sites consistently encompassed 7–13 barrels (Author response image 1, three exemplary injection sites). Additionally, we determined the centres of mass for each injection site and mapped them onto the barrel map. Four of the injection sites were located in the lateral part of SSp-bfd, two in the central region, and none in the medial part. Notably, the injection sites within SSp-bfd exhibited significant overlap. As a result, a selective analysis of callosal projections targeting these injection sites would likely not yield distinct projection patterns, as the projectomes would inevitably include projections to surrounding barrels, leading to contamination.

**Author response image 1. sa3fig1:** Left: exemplary Injection sites mapped onto the 3D barrel map of SSp-bfd within the Mouse Allen Brain Atlas. Barrels were reconstructed using a specialized software as described previously (Weiler et al., 2024) Right: Centres of mass of all SSp-bfd injection sites mapped onto the 3D barrel map.

Due to the fact we covered a significant proportion of the respective target primary sensory area any further subdivision of these data is not possible and requires more tailored injections into clearly defined subareas. Investigating the separate projectomes onto these subregions (e.g. onto V1M and V1B) remains an important interesting research question that we, at least in part, will address in a future study.

Finally, while the paper emphasizes that projections from L6 "dominate" intra and contralateral cortico-cortical inputs, the data shows a more nuanced scenario. While it is true that the areas for which L6 neurons are the most common source of cortico-cortical projections are the most abundant, the picture becomes less clear when considering the number of neurons sending these connections. In fact, inputs from L2/3 and L5 combined are more abundant than those from L6 (Figure 3B), challenging the view that projections from L6 dominate ipsi- and contralateral projecting cortico-cortical inputs.

We agree in the case of the barrel cortex, layer 5 significantly contributes in terms of the number of brain areas projecting from within the ipsilateral and contralateral hemispheres. Please see we have replaced the term “dominates” in the title, abstract and in the manuscript where relevant.

**Recommendations for the authors:**

**Reviewer #1 (Recommendations for the authors):**
The sections analyzing the role of L6 towards feedback (pg. 11-13, Figure 6) were a bit verbose and confusing to me. Three possible models are proposed:(1) a decrease in L23 projections, (2) an increase in L56 projections, or (3) both.However, what is being quantified appears to be the fractions inputs, with L23. L5, and L6 summing to 1. Thus, a decrease in L23 would necessarily result in an increase in L56 projections. It seems like it would make more sense to quantify the percent change in the total number of inputs (rather than fractional) from each layer so that the 3 models are actually independent possibilities.

The issue with the suggested analysis is that, with one exception (one area projecting to MOp), the number of projection neurons in contralateral areas is always ~60-80% lower compared to their ipsilateral counterparts. Consequently, this is also true for the number of projection neurons in the different cortical layers. Thus, quantifying the percentage change from the ipsilateral to the contralateral hemisphere in the total number of inputs from each layer will always result in negative values.

Nevertheless, we addressed the reviewer’s issue by calculating the preservation index (1(ipsi-contra)/(ipsi+contra)) for the sensory-motor areas independently for the absolute number of neurons within L2/3, 5 and 6 for the cortical areas projecting to VISp, SSp-bfd and MOp (see Author response image 2). When analysing the shift from the ipsilateral to the contralateral hemisphere, we observed that significantly more projection neurons were preserved in L6 compared to L2/3 for VISp and SSp-bfd. This shows that the number of L6 projection neurons declines less from the ipsilateral to the contralateral hemisphere compared to L2/3. However, our focus was on the fraction of projection neurons within each layer relative to the other layers per hemisphere (see Fig.6 of our manuscript). This measure is critical for distinguishing between feedforward and feedback connectivity. Calculating the change for each layer independently unfortunately does not provide insights into this comparison, as it does not capture the relative distribution of projection neurons across layers, which is central to our analysis. Therefore, we chose to present the data as layer fractions normalised within each hemisphere separately, enabling a comparison of relative changes between hemispheres, as shown in Fig.6 in the manuscript. We agree that with our approach a decrease in the fraction of L2/3 neurons would necessarily lead to an increase in the fraction of L5+6 neurons. However, as we analysed the fractional change for L5 and L6 separately, we found that the fraction of projection neurons in L5 generally showed only minor changes, while the fraction of L6 projection neurons increased substantially (Fig.6C). In addition, excluding L5 from the ipsi- or contralateral default network had significant effects on the fILN in only a relatively small number of projection areas. Excluding L6 resulted in significant changes in many more projection areas than layer 5.

**Author response image 2. sa3fig2:** Preservation index for L2/3, L5 and L6 of the 24 sensory-motor areas projecting onto the three target areas VISp, SSp-bfd and MOp.

**Reviewer #2 (Recommendations for the authors):**
I feel that there are a few conclusions that could be strengthened in the paper:(1) The laminar sources of callosal inputs and their terminations differ in the monocular and binocular areas of the visual cortex border with V2L. Similarly, callosal inputs are different close to the border of S1 with S2 than in the rest of the barrel cortex. From the methods sections and Figure S2, it seems that some injections targeted the V1 binocular zone while others were aimed at the monocular zone. Thus, it would be of interest to compare the laminar distribution and fILM of the contra inputs in inputs to the binocular and monocular zones (and S1 border vs the rest, if possible within this dataset).

Please see the answer for the reviewer’s second point in the public review (above).

(2) The results are currently a bit unclear on whether the contra inputs reflect the cortical hierarchy. Figure 4E-F makes it clear that the ipsi and contra fILMs do not always match. However, it seems from the plots in Figure 4D and Figure S6 that, while the contra fILM values are always higher, there might be a correlation between the ipsi and contra fILM. This could be addressed by directly plotting contra vs ipsi fILM.

Similarly, it would be useful to directly address if there is any hint of the visual hierarchy, as calculated in Figure S5 for the contra inputs.

Regarding the first point of the reviewer: We appreciate this comment. We do indeed find a positive correlation between the fILN ipsilateral and fILN contralateral across the individual cortical areas for all three targets. (please see Author response image 3 below). This is indeed an interesting observation that indicates a high degree of preservation concerning the rank order of the anatomical hierarchy within the input arising from both hemispheres. Please see we have included this new figure 4F into the manuscript and added a sentence in the results (lines 282-288):

Regarding the second point of the reviewer: For visual hierarchy, although weaker, we find that the hierarchical ranking of the higher cortical visual areas is preserved for the contralateral hemisphere (see Author response image 3 below).

**Author response image 3. sa3fig3:** Rank ordered average fILN values (± sem) of higher visual cortical areas of the ventral and dorsal visual stream for the ipsilateral and contralateral hemisphere.

(3) I find the emphasis in the title and other parts of the paper on Layer 6 corticocortical cells dominating the anatomical organization of intra and interhemispheric feedback a bit of an overstatement. While it is true that the areas for which L6 is the most abundant source of cortico-cortical projections are the most abundant (Figure 3C), when just focusing on the number of neurons sending corticocortical connections (Figure 3B), this is less clear. Ipsi connections are roughly divided 1/3, 1/3 , 1/3 between L2/3 , L5 and L6. In the contra, while projections from L6 neurons are the most abundant, there are not a majority and are less than those of L2/3 and L5 together. I suggest revising the statement about L6 cells dominating cortico-cortical connections to more accurately reflect these nuances.(4) The observations from Figure 3 discussed above suggest that L6 inputs dominate in areas with less abundant projections to the injected areas. Is this the case? Is the fraction of L6 inputs inversely correlated with the number of inputs from that area?

Please see the following correlation plots for the total number of inputs versus the fraction of L6 inputs per area for both the ipsilateral and contralateral hemisphere. We do find on the ipsilateral hemisphere a negative correlation between the total number of inputs and the L6 input fraction for VISp and to a lesser degree for SSp-bfd. Interestingly, we find the opposite correlation for the ipsilateral MOp, contralateral VISp, SSp-bfd and MOp (Author response image 4, Author response table 1). While this is an interesting finding, the correlations often appeared to be weak and often absent within the individual animals and across the three target areas (Author response table 1). Thus, these correlations are seemingly not a general feature of cortical connectivity.

**Author response image 4. sa3fig4:** Total number of cells versus fraction of cells within L6 per cortical brain area (average across animals) for the ipsilateral (top) and contralateral (bottom) hemisphere for the three target areas VISp, SSp-bfd and MOp.

**Author response table 1. sa3table1:** Respective correlations between total numbers of cells and fraction of cells within L6 per cortical brain area for the ipsilateral and contralateral hemisphere for the three target areas (significant correlations highlighted with green).

**VISp**	**ipsi**	
Animals	Correlation r Total numbers vs. L6 fraction
1	-0.6265
2	-0.6469
3	-0.2314
4	-0.6836
5	-0.6539
6	-0.4186
**contra**	
Animals	Total numbers vs. L6 fraction
1	0.196
2	0.398
3	0.646
4	0.1494
5	0.0579
6	0.3676
**SSp-bfd**	**ipsi**	
Animals	Correlation r Total numbers vs. L6 fraction
1	0.3942
2	0.0452
3	-0.0533
4	-0.4984
5	-0.234
6	0.0233
**contra**	
Animals	Correlation r Total numbers vs. L6 fraction
1	0.5548
2	0.2174
3	-0.0788
4	0.0369
5	0.0297
6	0.2001
**MOp**	**ipsi**	
Animals	Correlation r Total numbers vs. L6 fraction
1	0.3736
2	0.4007
3	0.4911
4	0.4287
5	0.1059
6	0.2606
**contra**	
Animals	Correlation r Total numbers vs. L6 fraction
1	0.5828
2	0.6672
3	0.6221
4	0.6355
5	0.5375
6	0.639

Minor issues:(5) Where was the mouse in Figure 3A injected?

In this exemplary mouse the retrograde tracer was injected into VISp. We added this information in the Figure legend of Figure 3A1.

(6) Clarify in panel 4F that the position of the circle corresponds to the area location.

Done as suggested.

References

Bieler M, Sieben K, Cichon N, Schildt S, Röder B, Hanganu-Opatz IL. 2017. Rate and Temporal Coding Convey Multisensory Information in Primary Sensory Cortices. eNeuro 4. doi:10.1523/ENEURO.0037-17.2017

Weiler S, Rahmati V, Isstas M, Wutke J, Stark AW, Franke C, Graf J, Geis C, Witte OW, Hübener M, Bolz J, Margrie TW, Holthoff K, Teichert M. 2024. A primary sensory cortical interareal feedforward inhibitory circuit for tacto-visual integration. Nat Commun 15:3081. doi:10.1038/s41467-024-47459-2

Yao S, Wang Q, Hirokawa KE, Ouellette B, Ahmed R, Bomben J, Brouner K, Casal L, Caldejon S, Cho A, Dotson NI, Daigle TL, Egdorf T, Enstrom R, Gary A, Gelfand E, Gorham M, Griffin F, Gu H, Hancock N, Howard R, Kuan L, Lambert S, Lee EK, Luviano J, Mace K, Maxwell M, Mortrud MT, Naeemi M, Nayan C, Ngo N-K, Nguyen T, North K, Ransford S, Ruiz A, Seid S, Swapp J, Taormina MJ, Wakeman W, Zhou T, Nicovich PR, Williford A, Potekhina L, McGraw M, Ng L, Groblewski PA, Tasic B, Mihalas S, Harris JA, Cetin A, Zeng H. 2023. A whole-brain monosynaptic input connectome to neuron classes in mouse visual cortex. Nat Neurosci 26:350–364. doi:10.1038/s41593-022-01219-x